# How youth cognitive and sociodemographic factors relate to the development of overweight and obesity in the UK and the USA: a prospective cross-cohort study of the National Child Development Study and National Longitudinal Study of Youth 1979

Drew M Altschul ,[1] Christina Wraw,[2] Catharine R Gale,[3] Ian J Deary[1]

¹Psychology, The University of Edinburgh, Edinburgh, UK
²National Health Service Scotland, Edinburgh, UK
³MRC Lifecourse Epidemiology Unit, University of Southampton, Southampton, UK

**Correspondence to**
Dr Drew M Altschul;
dmaltschul@gmail.com

## ABSTRACT

**Objectives** We investigated how youth cognitive and sociodemographic factors are associated with the aetiology of overweight and obesity. We examined both onset (who is at early risk for overweight and obesity) and development (who gains weight and when).

**Design** Prospective cohort study.

**Setting** We used data from the US National Longitudinal Study of Youth 1979 (NLSY) and the UK National Child Development Study (NCDS); most of both studies completed a cognitive function test in youth.

**Participants** 12 686 and 18 558 members of the NLSY and NCDS, respectively, with data on validated measures of youth cognitive function, youth socioeconomic disadvantage (eg, parental occupational class and time spent in school) and educational attainment. Height, weight and income data were available from across adulthood, from individuals' 20s into their 50s.

**Primary and secondary outcome measures** Body mass index (BMI) for four time points in adulthood. We modelled gain in BMI using latent growth curve models to capture linear and quadratic components of change in BMI over time.

**Results** Across cohorts, higher cognitive function was associated with lower overall BMI. In the UK, 1 SD higher score in cognitive function was associated with lower BMI (β=−0.20, 95% CI −0.33 to −0.06 kg/m²). In America, this was true only for women (β=−0.53, 95% CI −0.90 to −0.15 kg/m²), for whom higher cognitive function was associated with lower BMI. In British participants only, we found limited evidence for negative and positive associations, respectively, between education (β=−0.15, 95% CI −0.26 to −0.04 kg/m²) and socioeconomic disadvantage (β=0.33, 95% CI 0.23 to 0.43 kg/m²) and higher BMI. Overall, no cognitive or socioeconomic factors in youth were associated with longitudinal changes in BMI.

**Conclusions** While sociodemographic and particularly cognitive factors can explain some patterns in individuals' overall weight levels, differences in who gains weight in adulthood could not be explained by any of these factors.

## Strengths and limitations of this study

► This study used two independent cohorts from the USA and UK, each with more than 30 years of follow-up, allowing us to model levels and trajectories of body mass index cross-culturally.

► The cohorts are representative of the eras and nations they developed in, so our findings are limited to middle-aged Caucasian populations in Western nations.

► Cognitive and sociodemographic variables were available in both samples, and allowed us to compare the associations of these factors with the development and progression of overweight and obesity.

► More work is required to clarify if and how cognitive and sociodemographic factors are linked to overweight and obesity across the life course.

## INTRODUCTION

Obesity is increasingly prevalent worldwide,[1] doubling between 1980 and 2015 in 73 countries, among both children and adults.[2] Obesity and overweight are dangerously prevalent in both the USA and UK,[2] in 2015, more than 196 000 deaths in the USA and 26 000 deaths in the UK were attributable to overweight and obesity. Furthermore, obesity and overweight are linked to a range of physical ailments, including cardiovascular disease,[3] cancer,[4] type 2 diabetes and osteoarthritis,[5] as well as mental illnesses, such as depression,[6 7] anxiety[8] and bipolar disorder.[9] It is, thus, an important issue in public health to identify early-life factors that are associated with obesity and overweight in adulthood. To this end, we investigated the relationships

of sociodemographics and general cognitive function in youth with the onset and growth of obesity and overweight from the beginning of adulthood into middle age.

From the beginning of life, inherent biological factors such as sex play a role in determining bodily health,[10] quantified through measures like body mass index (BMI), which is used to define who is overweight and obese. BMI is estimated at between 40% and 70% heritable.[11] On the other hand, early-life socioeconomic disadvantage (SED) and adverse experiences are associated with higher risk of overweight and obesity,[12 13] though these findings are confounded by ethnic and geographical disparities.[14 15]

Cognitive function is also an important predictor of both physical and mental health outcomes,[16–21] and lower cognitive function is linked with obesity later in life.[22–24] Youth cognitive function in particular is the gold standard for cognitive epidemiological studies that seek to investigate the long-term associations between cognitive function and the development of disease; a measure of cognitive function from youth limits confounding. Cognitive function declines on average with age[25] and in the presence of disease comorbidity.[26–28] Moreover, obesity itself has been linked to high incidence of Alzheimer's,[29] Parkinson's[30] and other sources of cognitive decline and dementia in older age.[31] Therefore, early-life cognitive function measures are especially important for understanding the relationships between cognitive function and overweight and obesity.

Gradients in SED affect women's physical health differently from men's, such that greater SED in women is more consistently associated with poorer health.[32 33] Therefore, might gradients in cognitive function also affect women's health similarly, that is, with stronger associations between cognitive function and health than in men? In 1996, obesity was more likely in English women who were more disadvantaged, for example, had lower occupational status, but no such association was present in English men.[32] In related health conditions, such as diabetes,[33] heart disease[34 35] and hypertension,[36] women coming from greater disadvantage also had poorer disease outcomes. However, associations between early-life SED and health might be at least partially confounded by differential effects of early-life cognitive function, which are also associated with later life health outcomes, including overweight and obesity.[37] Cognitive function is an important predictor of later life health in both sexes, but it plays a stronger role in women. Evidence of this has been found in both the USA[38] and UK,[35 37] for high blood pressure, stroke and coronary heart disease.

Considerable longitudinal research is carried out in the USA and UK, but not nearly as much research has examined whether major correlates of illness are comparable between these nations. Our main goal in the present study was thus to compare, between both countries, whether cognitive function or SED in youth is associated with BMI from young adulthood to middle age. In particular, we wished to investigate the role of general cognitive function and its potential interaction with sex, in BMI changes from early adulthood to middle age. To this end, we analysed two large, comparable, longitudinal samples from the USA and UK, which allowed us to model growth curves and determine which early-life cognitive and sociodemographic factors are associated with higher BMI over the course of adulthood.

## METHODS

The American and British Samples are, respectively, the National Longitudinal Study of Youth 1979 (NLSY) and the National Child Development Study (NCDS). Both studies have followed participants from youth, when a test of general cognitive function was given. Sociodemographic variables from youth were also available, allowing us to consider other known factors that contribute to overweight and obesity. BMI was tracked longitudinally from participants' 20s into their 50s, specific details on the individual samples follow.

### The National Longitudinal Study of Youth 1979

The NLSY was initially sampled from non-institutionalised young Americans born between 1956 and 1964; aged 14–21 years and living in the USA at the beginning of 1979. The original sample consisted of 12 686 participants, and was ethnically representative of the USA at that time; 16% of participants were 'Hispanic', 25% were 'black' and 59% were 'non-black non-Hispanic'; a total of 7506 individuals were in this last category, so 5172 NLSY participants were either black or Hispanic. The initial interview took place in 1979, and respondents were regularly reinterviewed up to 2014, which is when the most recent data were available. A total of 3396 individuals participated in giving height and weight measurements in 2014 (table 1). The ethnic categories were less specific than in the NCDS, so for example, some Asian-Americans were likely included as 'non-black non-Hispanic', though in 1980 only 1.5% of the US population was of Asian or Pacific Islander descent.

### The National Child Development Study

Participants of the NCDS were born in the UK during 1 week in March 1958. As originally intended, the NCDS was designed to study stillbirth and death in infancy. Later developments in the survey have resulted in multiple follow-ups at regular intervals. Eleven waves of data have been collected, the first at birth and the most recent in 2013, when participants were 55 years old. Of the initial 18 558 participants, 12 440 (67%) were 'Euro-Caucasian'. By the 2013 wave, 6861 (37%) individuals participated in giving height and weight measurements, again see table 1. Analyses were limited to only non-minority ethnic group individuals to allow between-country comparisons.[39 40]

### Exposures

In the NLSY, general cognitive function was assessed using the Armed Forces Qualification Test (AFQT), 1989 renormed version. The AFQT consisted of four subtests

**Table 1** Descriptive statistics for all tested variables

| Variable | NLSY—USA | | | | NCDS—UK | | | |
|---|---|---|---|---|---|---|---|---|
| | Average age | N | Median | SD | Average age | N | Median | SD |
| BMI (kg/m²) | 24 | 6198 | 22.71 | 4.00 | 23 | 9835 | 22.13 | 3.09 |
| | 33 | 4254 | 24.96 | 4.98 | 33 | 8741 | 24.62 | 4.57 |
| | 45 | 3651 | 26.62 | 5.53 | 42 | 8787 | 25.43 | 4.66 |
| | 53 | 3396 | 27.77 | 6.04 | 55 | 6861 | 26.79 | 5.49 |
| Net family income (US$ or £) | 24 | 5386 | US$382 | US$365.17 | 23 | 10027 | £87 | £52.25 |
| | 33 | 3709 | US$769 | US$766.83 | 33 | 9094 | £272 | £178.64 |
| | 45 | 3593 | US$1288 | US$1624.87 | 42 | 9032 | £435 | £399.02 |
| | 53 | 3092 | US$1415 | US$2133.50 | 50 | 7846 | £600 | £554.46 |
| Cognitive function | | 7023 | 0.12 | | | 11571 | 0.13 | |
| Youth SED | | 7023 | 0.01 | | | 12290 | 0.15 | |
| Education | | 3788 | −0.03 | | | 8819 | −0.53 | |
| Sex | | | 3787 male | 3719 female | | | 5633 male | 5320 female |

Cognitive function, youth SED and education are unitless variables, thus they are scaled so that their SD is 1. The means are 0, so the medians indicate the amount of skewness in the distributions.
BMI, body mass index; NCDS, National Child Development Study; NLSY, National Longitudinal Study of Youth 1979; SED, Socioeconomic Disadvantage.

assessing arithmetic reasoning, mathematical knowledge, word knowledge and paragraph comprehension. The test was given in 1980, when participants were between 15 and 22 years old. The AFQT is a valid and reliable measure of general cognitive function, having been associated with outcomes including academic achievement and job performance.[41 42] To be consistent with earlier work,[20 43] including The Bell Curve,[44] we used the z-scored AFQT percentile score in our analyses. In the NCDS, cognitive function was assessed using a general ability test, given to participants at age 11.[45] The test consisted of 40 verbal and 40 non-verbal items, and the total score was z-scored for analysis. The reliability is high (test–retest Cronbach's α=0.94), and the test is valid, as indicated by its high correlation (r=0.69–0.93) with tests used for secondary school selection.[46]

Due to historic data collection limitation, youth SED was composed of differently in each sample, but both measures have been previously validated. In the USA, youth SES was the sum of z-scored variables for parental income, education and occupation status. Like AFQT scores, this variable has been previously used and validated in previous work.[43 44] In the UK, youth SED was the sum of six z-scored variables: father's social class at birth, father's social class at age 7, age at which the father left education, age at which the mother left education, parental housing tenure in childhood and the number of people sharing a room in the household at age 7. This variable was composed of to be consistent with earlier work.[47]

## Outcome

BMI was assessed 4 times, when the average age was 24, 33, 45 and 53 in the NLSY, and 23, 33, 42 and 55 in the

NCDS. BMI was calculated as weight divided by the square of height, and where measurements were taken in Imperial units, they were converted to metric. Values of BMI greater than 70 or less than 12 were treated as biologically implausible[48] and the observations were removed from the dataset. From the NCDS, 0 such cases needed to be removed at age 23, 65 at age 33, 11 at age 42 and 4 at age 55. From the NLSY, only two such cases needed to be removed, at the 1985 wave.

## Covariates

Education was measured in age at which an individual left school in the NCDS, and highest grade achieved in the NLSY, which was only converted to age at which an individual left school for descriptive purposes in table 1. Each variable was independently z-scored for use in our models.

Net family income was used in both samples. Net family income was derived from a comprehensive set of income questions in both samples. The possible sources came from all earning members of the household, and included military income, wages, salaries, tips, unemployment compensation, child support, alimony, food stamps, welfare and disability benefits, interest and dividends and others. The derivation of these variables in the NLSY is documented on the National Longitudinal Surveys website (https://www.nlsinfo.org/content/cohorts/nlsy79/topical-guide/income/income/) and in the NCDS the derivation of these variables was managed by the Centre for Longitudinal Studies[49] (see also online supplementary appendix 1). Both income variables were top coded to ensure confidentiality of each sample's top

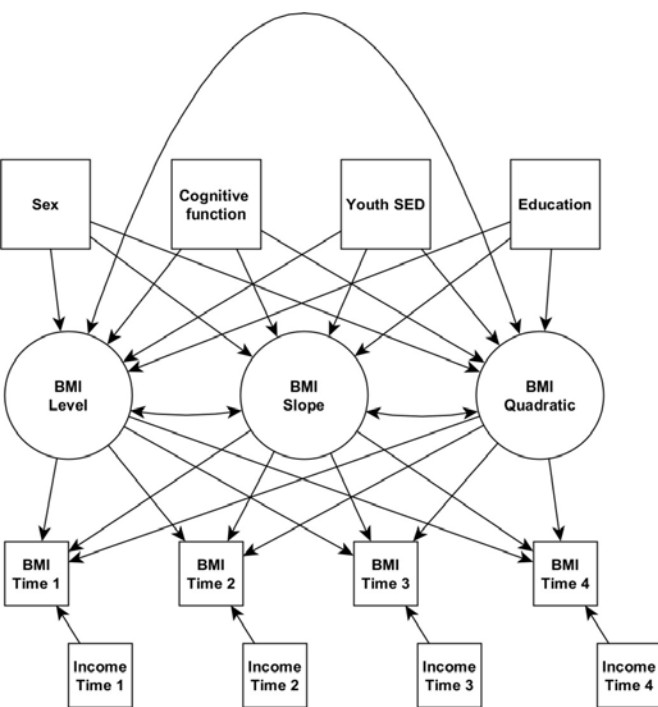

**Figure 1** Path diagram of latent growth curve models used in primary analyses. Square boxes are observed variables, circles are latent variables. Single headed arrows are regressions or latent variable loadings, double headed arrows are covariances. BMI, body mass index; SED, socioeconomic disadvantage.

earners, which also protects against some of the effects of large outlying observations.

### Latent growth curve modelling

We modelled change in BMI across time using latent growth curve models (LGCMs). LGCMs are a form of structural equation model[50] where, in this case, the growth curve model fits latent variables to observed variables, with the latent variables representing an individual's overall level (intercept) of a variable across time, as well as latent variables for change (slope or trajectory) over time. These latent variables can be related to other observed variables with conventional regression and covariance methods.

The latent variables in our model were derived from four BMI measurements taken at roughly decade-long intervals. The BMI latent variables included variables for overall BMI level (or intercept), linear change and quadratic change. All analyses were carried out using the R package 'lavaan'.[51]

Each BMI latent variable was regressed on youth SED, education, sex, cognitive function, and the interaction between sex and cognitive function. BMI measurements were regressed on income measurements corresponding to the same measurement time points. Because there was substantially more age variation among NLSY participants, additional control variables were included in those models. These were age at cognitive testing in 1979, on which was regressed the BMI latent variables, as we did

with cognitive function and other time-invariant variables, and age during each follow-up wave, on which, like income, was regressed the BMI measurements. The variances of the first BMI observations had to be constrained to be greater than 0 in both the NLSY and NCDS models, but this did not substantially impact the fit of either model.

All p values are two sided and have been corrected for multiple comparisons using the False Discovery Rate[52] within models. Missing data handling procedures are described in online supplementary appendix 2 of the supplementary information.

### Patient and public involvement

Data were deidentified and no patients were involved in this research. The results of this project will be disseminated via the usual academic media and in the press.

### RESULTS

### Overview of the results

We first describe the ultimate structure of our LGCMs, second we describe the general appearance of the results, and then report the results of the models in each country separately. All models were a variation on the one presented in figure 1. BMI change over time was the best modelled with quadratic and linear slope components in both samples (NLSY: $\chi^2$=30.743, df=13, CFI=0.998, SRMR=0.009; NCDS: $\chi^2$=286.905, df=1, CFI=0.985, SRMR=0.027), suggesting that BMI increases over time, and that the rate of increase generally decreases over time (online supplementary table S1). The main results of our models are shown in table 2, and explained below.

In both the USA and the UK, and in men and women, mean BMI increases from the 20s to the 50s, by about 4–5 kg/m² (table 1, figure 2). Men appear to have higher mean BMI than women (figure 2, upper panel). Americans have higher BMI than the British from early adulthood, and the differences grow as adulthood advances; Americans were more than half a BMI unit higher than the British when participants were in their 20s, and nearly 1.5 BMI unit higher when participants were in their 40s. Although cognitive function was modelled as a continuum, we show it in figure 2 as tertiles for descriptive purposes only. In both the USA and the UK, throughout the period of adulthood examined, the highest cognitive function tertile has the lowest mean BMI and the lowest tertile has the highest BMI (figure 2, middle panel). This was seen in both countries, and in men and women (figure 2, lower panel). Next, we report the results of formal modelling of that data.

### USA

In the US sample, men had higher BMI level than women, but there were no main effects of cognitive function, early-life sociodemographic characteristics or education (table 2). There was evidence for an interaction between sex and cognitive function on BMI; the model's results

**Table 2** Coefficient estimates from models of BMI growth, cognitive function and sociodemographic variables

| Outcome | Predictor | USA | | | UK | | |
|---|---|---|---|---|---|---|---|
| | | Estimate | 95% CI | P value | Estimate | 95% CI | P value |
| BMI level | Sex (women) | **−1.444** | **(−1.820 to −1.068)** | **<0.001** | **−0.849** | **(−1.016 to −0.681)** | **<0.001** |
| | Cognitive function | 0.105 | (−0.221 to 0.431) | 0.683 | **−0.196** | **(−0.330 to −0.063)** | **0.011** |
| | Sex x cognitive function | **−0.525** | **(−0.901 to −0.149)** | **0.046** | −0.053 | (−0.222 to 0.116) | 0.694 |
| | Youth SED | 0.178 | (−0.047 to 0.403) | 0.348 | **0.332** | **(0.230 to 0.434)** | **<0.001** |
| | Education | −0.248 | (−0.497 to 0.001) | 0.225 | **−0.153** | **(−0.263 to −0.042)** | **0.017** |
| BMI slope | Sex (women) | 0.157 | (−0.230 to 0.544) | 0.670 | **−0.327** | **(−0.504 to −0.150)** | **0.001** |
| | Cognitive function | 0.092 | (−0.254 to 0.438) | 0.733 | 0.036 | (−0.110 to 0.182) | 0.694 |
| | Sex x cognitive function | 0.295 | (−0.087 to 0.677) | 0.348 | −0.140 | (−0.322 to 0.041) | 0.239 |
| | Youth SED | 0.281 | (0.045 to 0.518) | 0.108 | 0.028 | (−0.084 to 0.140) | 0.694 |
| | Education | −0.082 | (−0.332 to 0.168) | 0.683 | 0.030 | (−0.082 to 0.143) | 0.694 |
| | BMI level | −0.319 | (−0.810 to 0.171) | 0.443 | **0.384** | **(0.184 to 0.584)** | **0.001** |
| BMI quadratic | Sex (women) | 0.043 | (−0.085 to 0.172) | 0.683 | **0.135** | **(0.080 to 0.189)** | **<0.001** |
| | Cognitive function | −0.003 | (−0.119 to 0.113) | 0.978 | −0.009 | (−0.054 to 0.037) | 0.742 |
| | Sex x cognitive function | −0.095 | (−0.222 to 0.032) | 0.348 | 0.026 | (−0.031 to 0.084) | 0.541 |
| | Youth SED | −0.038 | (−0.118 to 0.043) | 0.627 | 0.027 | (−0.009 to 0.062) | 0.242 |
| | Education | 0.020 | (−0.063 to 0.104) | 0.733 | −0.015 | (−0.051 to 0.021) | 0.563 |
| | BMI level | −0.002 | (−0.166 to 0.161) | 0.978 | −0.071 | (−0.139 to −0.003) | 0.083 |
| | BMI slope | **−1.554** | **(−1.810 to −1.298)** | **<0.001** | **−1.006** | **(−1.146 to −0.866)** | **<0.001** |
| $\chi^2$ | | 83.966 | | | 137.465 | | |
| df | | 31 | | | 18 | | |
| CFI | | 0.992 | | | 0.991 | | |
| SRMR | | 0.015 | | | 0.011 | | |

Models are latent growth curve models, with latent variables for BMI level, slope and quadratic slope—see the path diagram in figure 1. The same path diagram was used for both samples, except the US sample was modelled with adjustments for individual ages. All coeffecient values are multiple regression coefficients from the predictor onto the outcome, except the associations between the three BMI latent variables, which are covariances. Effect size estimates are for 1 SD changes in a variable. P values are corrected for multiple comparisons using the false discovery rate. Bolding indicates estimates with P values < 0.05.
BMI, body mass index; SED, socioeconomic disadvantage.

suggest that cognitive function is not related to BMI level in American men, but, in American women, 1 SD higher cognitive function is associated with a more than half unit lower BMI level.

We found no associations between early-life cognitive function or sociodemographic variables and rate of BMI increase, although linear BMI change was negatively associated with quadratic BMI change. This suggests that more rapid BMI increases earlier in life are accompanied by a flattening-out of BMI change in middle age, which is consistent with what we see in our visualisations (figure 2).

## UK

In the British sample, women had lower BMI levels than men, more than 0.8 units all else being equal (table 2). Higher cognitive function in youth was associated with lower BMI level; 1 SD higher score in cognitive function was associated with ~0.2 units lower BMI. There was no sex by cognitive function interaction on BMI level. Additionally, and unlike in the American sample, high youth SED and more educational attainment were associated with higher and lower BMI levels, respectively. We note

that these coefficients in the US samples are not dissimilar to those in the UK sample, and that the UK sample is larger and probably has greater power.

As found in the American sample, there were no relationships between BMI growth and cognitive function, youth SED or education. However, there was an association between BMI growth and sex, so that women had slower growth in BMI over time. Every decade, men gained about 0.3 BMI units more than women, all else being equal. Sex was also associated with the quadratic component of BMI growth, such that men's BMI growth tended to level off in later ages, whereas women's linear BMI growth continued more linearly in later ages. These British effects are visible in figure 2.

In the UK sample, BMI level was associated with linear BMI slope, and linear slope was associated with quadratic slope. This means that higher BMI level indicated greater linear growth, and having a higher linear growth component was associated with more flattening out of growth over time. At one point in time, and only in the British sample, income was related to BMI: at age 23, having a

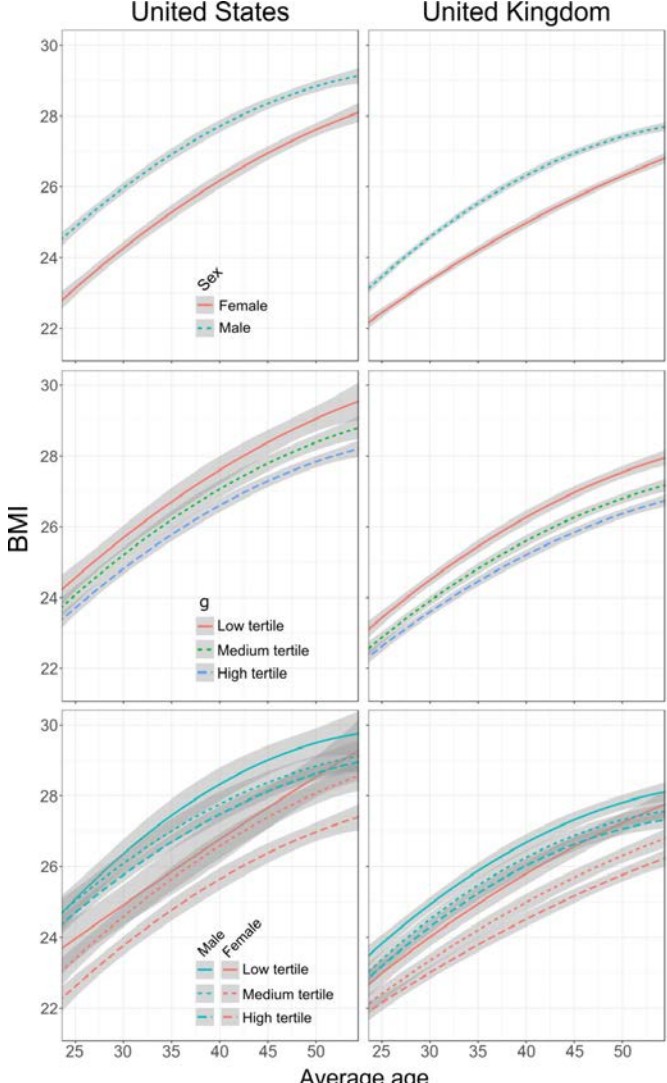

**Figure 2** Quadratic growth curves of BMI fitted to tertiles of cognitive function and across different sexes. The top panels illustrate BMI between sexes, the middle panels illustrate BMI in different general cognitive function (g) tertiles and the bottom panels illustrate the interaction between cognitive function tertiles and sex. Shaded areas represent the 95% error regions. BMI, body mass index.

higher income was associated with having higher BMI at that age (B=0.574, 95% CI 0.390 to 0.758, p<0.001).

### Sensitivity analyses

As our primary interest was in early-life factors, we wished to explore how removing other variables, first income, then education, might influence associations with BMI variables. Moreover, some of these income and education measurements are missing for some participants, so fitting models without them would give us additional power to detect effects, by not requiring that our models rely as heavily on maximum likelihood estimation to fill in the missing values.

Modelling without income (online supplementary table S2) allowed us to confirm the association between higher education and lower BMI levels in the USA (B=−0.339,

95% CI −0.534 to 0.143, p=0.004). Associations were unchanged in the UK when income was removed. Going further and removing education (online supplementary table S3) revealed associations in the USA between SED in youth and BMI level (B=0.274, 95% CI 0.101 to 0.448, p=0.010) and BMI slope (B=0.231, 95% CI 0.051 to 0.412, p=0.030), such that more disadvantaged Americans have higher overall BMI levels and faster growth. Removing education in the UK model revealed an association between youth SED and quadratic BMI growth (B=0.032, 95% CI 0.004 to 0.060, p=0.045), which would suggest that all else being equal, more disadvantaged British participants showed less levelling off of BMI growth over time.

We also wished to see if our findings generalised to the entire populations represented in these samples, so we refit our primary models (ie, those described in table 2) with all participants from each sample, regardless of their ethnic group. All of the effects previously described persisted with similar effect sizes (online supplementary table S4). Cognitive function was associated with higher BMI level across sexes in the UK, but the protective effect of cognitive function was only present in American women.

### DISCUSSION

From their 20s to their 50s, both the US and UK participants had similar starting points and trajectories of growth in BMI. Almost everyone gained weight as they aged. Both populations show steeper BMI growth in their 20s and 30s, and neither sociodemographic nor cognitive factors were associated with these changes. Nevertheless, some factors did predict whether participants would have lower or higher BMI levels that would be reflected from early adulthood into middle age, though these factors often differed between the American and the British samples.

A straightforward source of difference in BMI levels came from sex. Both American and British women have lower BMIs than men. However, American women are likely to have nearly 1.5 fewer BMI units than American men, whereas British women are likely to have only about 0.85 fewer BMI units than British men. A possible reason is visible in figure 2: Americans weigh more in general and that allows Americans to have greater range of BMI values, so for example, there is more opportunity for American men to have higher BMI than the average American.

In their 20s, the lowest BMI American women approximately matched the lowest BMI British women, but the American sample rapidly gained more weight, and by their 40s, the average American is more than a full BMI unit higher. Diversity in BMI grows across both samples as well: the SD of BMI among Americans goes from four units in the early 20s to more than six in the 50s, and among the British the SD rises from ~3 at age 23 to ~5.5 at age 55. BMI growth is faster in British men than British women, but that growth slowed in participants' 40s and

50s, whereas in British women, growth was slower but more consistent. No such effects were apparent in Americans. Both men and women tended to gain weight at the same rate.

Associations between cognitive function and BMI levels were present in both samples. British with higher cognitive function tended to have lower BMI: a British person of average height (~1.7 m) with a 15 IQ point, which is equivalent to 1 SD, lower score than average would likely weigh at least half a kilogram (1.1 lbs.) more than average. On the other hand, only higher cognitive function American women were likely to have lower BMI. For an average American woman (~1.6 m), a 15 IQ point (again, 1 SD) lower score than average would likely be equated with being 1.3 kg heavier (2.9 lbs.). Cognitive function was one of the more noticeable sources of difference in BMI (figure 2); in the American sample, only cognitive function and sex, not socioeconomic factors, appeared to be associated with BMI.

In addition to cognitive function, education and youth SED were also associated with BMI levels in British people, but not in Americans. Having more education was associated with lower BMI, to nearly the same extent as cognitive function. Coming from a less deprived background was also associated with lower BMI, to a greater degree than either cognitive function or education, though, again, only in the British. In the American sample, education showed a similar effect size and a nearly overlapping CI; therefore, with more participants and thus power, we might have found that education was similarly associated in the American sample. Some, but not all, of our sensitivity analyses support this, so imprecision of variable measurement might also explain the width of this CI. Income was not associated with BMI at any point in our analyses, except among the British when they were age 23. At this age, income may be related to BMI in the same way as youth SED, which is composed of parents' sociodemographic variables from a few years earlier.

Particular strengths of this study include that it is longitudinal, with multiple measurements spanning 30 years. Moreover, a valuable and unusual strength is that the variables and their life-course timings in the two countries were remarkably similar. We identified and derived highly analogous variables for every major factor in the study, making the direct comparisons across these two cultures feasible.

There were some limitations to this study as well. For greater comparability between samples, we focused our study on American and British participants of European descent. However, our own sensitivity analyses as well as previous analyses give us no reason to believe that these findings are not applicable in other ethnic groups.[19 38] The NLSY and NCDS samples are now both middle aged, so our growth curve models may not fit in precisely the same way in other age groups, particularly younger samples, whose early-life background may have been different from what our participants experienced in the 1950s, 60s and 70s. Additionally, our American sample

was smaller than our British sample, so we did not have as much power to detect relationships in Americans.

BMI is an imperfect measure[53]; it does not always effectively capture individual adiposity, nor does it indicate that any particular mechanism is at work. However, we did not have access to repeated measures of other adiposity measures, for example, waist circumference or per cent body fat. Nevertheless, BMI is important to study as it remains and will continue to be widely used, and it is a risk factor for many illnesses.[3–6]

Previous work with the NLSY has demonstrated that trajectories of BMI over the same stretches of adulthood are associated with social factors including sex, SED and education.[54 55] Cognitive function in youth has also been linked to maternal and offspring BMI in the NLSY,[20] but to the best of our knowledge, cognitive function has not been associated with trajectories of BMI across the life course. Similarly, studies with British cohorts, including the NCDS, have found that socioeconomic disparities are associated with BMI differences,[56 57] but cognitive function and trajectories of BMI are much less studied. A notable exception is Chandola et al[37] who also used LGCMs with the NCDS, but analysed men and women in different models, thus could not detect the interaction effects we found. Our use of LGCMs allowed us to model change in individuals' BMI over the same period of time, incorporating the same variables, but in distinct cultures. Moreover, our analyses being cross-cultural provide further validation for the effects we found that spanned both of our samples, thus demonstrating that these results are not unique to a particular nation or culture.

What is apparent is from our results is that cognitive function tested in youth has an important association with BMI, from early adulthood into middle age. BMI and cognitive function are genetically correlated at $r_g = -0.13$,[58] so some of the association we found might be caused by genetic variation; however, some of the genetic risk of obesity can be reduced through education.[59] It is; therefore, important to ask: do people learn things over the course of their education that they use to live healthier lives? In much the same way, we might be able to encourage what has been termed 'phenocopying'[17]—that is, encouraging and enabling people to follow the same strategies that individuals with higher cognitive function use to look after their health with the hope of achieving the same results. In contrast to education, which we only found to be associated with BMI in the UK, we found associations between cognitive function and BMI in both samples, as did two prior studies.[58 59]

The associations between cognitive function, sociodemographic factors and BMI were mostly with an individual's overall BMI level. BMI level is thus of particular importance because it indicates whether a person is overweight or obese from the start of these samples, and obesity among younger adults is particularly hazardous to one's later life health.[60] Youth SED might be linked to major early-life determinants of overweight and obesity,[61] and so, through phenocopying and related health education,

early interventions are probably best positioned to make a difference to individuals' lifelong health.

This study shows that BMI growth is similar across two nationalities, in spite of major differences in BMI baselines between the USA and UK. Moreover, BMI growth appears to be largely unaffected by cognitive and socio-demographic factors. Baseline BMI levels (rather than their rate of change) are influenced by external factors—cognitive function being one of the strongest; its effect appeared in both the USA and UK samples. Identifying the behaviours and other pathways through which cognitive function might protect individuals from overweight and obesity should be investigated further in the pursuit of real-world solutions that can help us to reduce the global burden of the overweight and obesity epidemic.

**Contributors** DMA discussed and planned the study and analyses, analysed the data, interpreted the data and drafted the initial manuscript. CW discussed and planned the study and analyses, interpreted the data and contributed to the manuscript. CRG discussed and planned the study and analyses, interpreted the data and contributed to the manuscript. IJD discussed and planned the study and analyses, interpreted the data and contributed to the manuscript.

**Funding** This work was supported by the University of Edinburgh Centre for Cognitive Ageing and Cognitive Epidemiology, part of the cross council Lifelong Health and WellbeingWell-being Initiative (MR/K026992/1). Funding from the Biotechnology and Biological Sciences Research Council (BBSRC), Economic and Social Research Council (ESRC) and Medical Research Council (MRC) is gratefully acknowledged. This work was also supported by an MRC Mental Health Data Pathfinder award (MC_PC_17209).

**Competing interests** None declared.

**Patient consent for publication** Not required.

**Provenance and peer review** Not commissioned; externally peer reviewed.

**Data availability statement** Data may be obtained from a third party and are not publicly available.

**ORCID iD**
Drew M Altschul http://orcid.org/0000-0001-7053-4209

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
