## [Reviewer comments · BMJ Open]

ARTICLE DETAILS

TITLE (PROVISIONAL)	How youth cognitive and sociodemographic factors relate to the development of overweight and obesity in the UK and the US: a prospective cross-cohort study of the National Child Development Study and National Longitudinal Study of Youth 1979
AUTHORS	Altschul, Drew; Wraw, Christina; Gale, Catharine; Deary, Ian

VERSION 1 – REVIEW

REVIEWER	Leah Li UCL Institute of Child Health
REVIEW RETURNED	12-Aug-2019

GENERAL COMMENTS	The authors investigated the associations of cognitive and sociodemographic factors in early-life with trajectories of adult BMI over three decades in the US National Longitudinal Study of Youth 1979 (NLSY) and the UK National Child Development Study (NCDS). I have several points for the authors: 1. The introduction could be shortened and focus more on the research question. There is little mention on why they conducted this research using data from these two countries.2. Page 8 lines 152-53: 'BMI and income were assessed at 4 times...' should be 'BMI was assessed at 4 times...'3. Latent Growth Curve Modeling – line 163 and elsewhere, the authors stated 'The BMI latent variables included variables for overall BMI level, linear slope, and quadratic slope'. I assume the authors meant intercept, slope and the coefficient for the quadratic term of age for BMI. It is not clear what age the overall BMI level is referred to.4. Line 165-66: 'Youth SED, education, sex, cognitive function, and the interaction between sex and 166 cognitive function were regressed on each of these BMI latent variables'. In fact BMI was regressed on Youth SED, education, sex, cognitive function, not the other way around.5. Page 195-199: How did the authors conclude that the rate of BMI increase decreased over time? This can only be obtained from the formula of the growth curve (i.e. the coefficient for the quadratic age term is negative).6. Page 210: '4 to 5 BMI points' should be '4 to 5 kg/m²'.7. A general point: for a maximum of 4 BMI measurements per person in the two cohorts, the authors used a very complicated model, with many latent variables – i.e. random intercept, random slope, and random coefficient for age², as well interaction of each factor (e.g. sex, cognition, SEP, education) with these random effects, resulted in a large number of estimates for coefficients and variance/covariance. In table 2, the authors included all the
---

	interaction terms with age and age2 that are not significant. If they exclude all the interactions with age2 that are not significant from the model, some interaction terms with age might become significant. Similarly, if they further remove the interactions with age, some main effects might be significant. Also for these interactions, fixed (not random) effects for age and age2 might be sufficient and therefore the models would a lot simpler and findings might be different. This is only a suggestion.
--	--

REVIEWER	Veronique Gingras Postdoctoral fellow, Harvard Pilgrim Health Care Institute and Harvard Medical School, United States
REVIEW RETURNED	11-Sep-2019

GENERAL COMMENTS	Overall comments This is an interesting study with a novel research question. The methods used to assess BMI trajectories are also interesting. Yet, there are some limitations for the interpretation of these findings and clarifications are needed. First, the authors state in the title and throughout the manuscript that they are looking at early-life cognitive and sociodemographic factors and their relationship with overweight and obesity. However, cognitive function was assessed between 15 – 22 years in one cohort, and at age 11 in the second cohort. SED appears to have been measured in adolescence or young adulthood as well? Perhaps it would be more adequate to change early-life for adolescence or youth? Also, there is missing information in the methods, and it makes this study harder to interpret. There seem to be many differences between the two cohorts, and it is unclear whether they can really be compared. For example, the authors give very little information regarding the cognitive function test, which is not the same in both cohorts, and was not assessed at the same time. This is the main exposure variable, I think it would be best to include it in the main manuscript rather than in an appendix, it would be easier to read. Same for education level and SED which are not assessed the same way in both cohorts. Also, is cognitive function assessed at 11 vs. 15-22-year-old comparable? How does this difference affect the results? Authors refer to IQ in two places in the manuscript, was there a specific IQ test performed or does this refer to the cognitive test results? Specific comments Title: See previous comment regarding “early-life” Abstract: Lines 32 – 33: What do authors mean by “most of both studies completed a cognitive function test in youth”? The information regarding the exposure variable (how it was assessed, when and in how many participants) should maybe be part of a sub-section “exposure”? Line 36: Consider replacing “early-life” and mention the age range when it was measured Line 36: Socioeconomic disadvantage: Could authors briefly define? Line 37: Adulthood – specify which period Line 43: change “cultures” for “cohorts”
--

	Line 44: The part of the sentence “a ~0.2” could be removed since we have this information with the β coefficient Lines 44-45: With the β coefficients, add the unit (kg/m²): -0.20, 95% CI -0.33; -0.06 kg/m²; also, the p values are not needed considering we have the CI which doesn’t cross the null; Same for the results on lines 46, 49, 50 Line 47: change “more than half a point lower BMI” for “lower BMI”, we already have the value with the β coefficient. Lines 50 – 51: Add the “timeline”: Overall, no cognitive or socioeconomic factors in youth were associated with longitudinal changes in BMI throughout adulthood. Lines 53 – 55: Perhaps the conclusion should be focused on the primary outcome? What do authors mean by “some average weight pattern”? Strengths and limitations: Consider mentioning differences in assessment methods Line 67: Is “cross-culturally” appropriate here? Change for “in two different populations”? Lines 73-74: Considering the null results, is it appropriate to mention mechanisms? Maybe change to mention that more studies are needed to clarify if and how cognitive function and SED are linked to obesity Introduction: Overall, the introduction is quite long. For example, the sentence from lines 80 – 83 might be unnecessary. It is known that obesity is related to poor health outcomes, and this study focuses on BMI itself, not associated co-morbidities. Line 94: Is it really established that it “precedes obesity”? Has the causal effect been studied, or would it be more appropriate to mention that cognitive function has been found to be linked to later obesity? Lines 97-98: It is probably an overstatement to say that using an earlier cognitive function measurement avoids confounding. Consider changing for limits confounding and consider moving this sentence to the discussion. Also, can authors be sure that their measurement is pre-morbid? Did authors exclude from their analysis participants with obesity or complications at the cognitive function assessment? Lines 103-115: This paragraph is quite long and maybe not all information is needed to introduce this study – some of the information in this paragraph would be more appropriate for the discussion Lines 116-117: “major early-life influences”: change for the specific exposure in this analysis Lines 120 – 127: Move this information to the methods? Methods: Line 134: What does “- 7506 individuals” represent? Line 137-140: Is this information needed? Line 147: Give proportions instead of numbers For both the NLSY and the NCDS studies, specify how many participants were included for this analysis. A flow-chart with participants number (baseline and different assessments) in the original cohort vs. included in this study would be helpful The methods section would be easier to read if reorganized: before variables, authors could describe the exposures, then outcomes, then covariates, and finally statistical analyses. Also, the information in this section is insufficient; it is not ideal to have
--	---

	to go through supplementary material to understand what the exposure is and how it was assessed. Most of the information in Appendix 1 should be in the main manuscript. There are several other sections where authors could shorten or rephrase the information to leave more room for the methods. Results: Lines 183 – 187: This section is difficult to follow. Ethnicity was not available for all participants? Give proportions instead of numbers. Minority ethnic groups were excluded from the analysis? That was not mentioned in the methods and could possibly introduce selection bias. What does this exclusion represent in terms of # of participants? The flow-chart would be helpful to clarify this. Lines 190 – 194: This information would be best suited for the methods section. Lines 195 – 205: Again, most of this information belongs with the methods. Consider clarifying the paragraph on BMI trajectories in the methods, and, if needed, adding an appendix with non-essential information regarding the methods for trajectories. Lines 210 – 215: consider changing BMI point for BMI unit – same for throughout the manuscript Lines 215 – 219: Was IQ the measure of cognitive function? Through the results, authors use “BMI growth”; yet, growth usually refers to height. Consider changing for BMI trajectory, or rate of BMI increase where appropriate. Lines 240 – 242: What do authors mean by the UK sample had more power? Were calculations performed or are they referring to the # of participants alone? Is it possible that the measurements were more or less precise in one of the cohorts? Also, this information would be best suited for the discussion. Discussion: Overall, the discussion is quite long and could be focused. A lot of the information in the discussion is simply a summary of the findings rather than the interpretation of the observed associations, or lack of associations. Line 279: Change “appeared to prevent this” for “were associated with these changes”. Lines 285-288: Differences in BMI by sex was not the primary outcome of this study; perhaps this section should be shortened into one sentence mentioning that in this study, most of the change in BMI over time was explained by sex? Lines 287-288: What do authors mean? Lines 289 – 297: Authors discuss differences in BMI change from the US vs the UK cohort in the results, and again in the discussion; yet, is this a primary focus of this study? Shouldn't the focus be how cognitive function and SED affect BMI differentially in both cohorts? Also, how comparable are the cohorts beyond BMI? There are not that many baseline variables to understand how different the cohorts are. Lines 298 to 306: IQ or cognitive function? Why present the difference for 15 points in both cohorts when the tests and probably the scales are not the same? Maybe present what 1SD change in cognitive function represents in terms of weight in both cohorts, and specify how many points this 1SD change represents? Lines 311 – 314: With more participants? The # of participants is already quite high. Could it rather be the precision of the tool that lead to a larger CI?
--	--

	Lines 323-325: As previously mentioned, this was not indicated in the methods. Why include only participants of European descent? Authors should consider/discuss how this exclusion could introduce selection bias and sensitivity analyses including all participants should be presented. Lines 325 – 331: This is unclear. Why do authors believe that their results could be applicable to other ethnic groups if only European descent participants are included? Lines 331-333: The number of participants is already quite large, see previous comment regarding precision of the tools vs. # of participants for CI precision Lines 334 – 335: At which time points? Different assessment timing from the current analysis? Lines 344 – 345: Even though results were different in the two cohorts? Lines 346 – 356: I do not quite understand this paragraph. Why are authors discussing genetic associations? Isn't this paragraph out of the scope of this analysis? Lines 357 – 363: The fact that associations were only found with overall BMI and not BMI change over time does not really support the importance of overall BMI. BMI is not a perfect method to assess obesity. Several studies showed that BMI does not always reflect adiposity and cardiometabolic risk, authors could mention this limitation of this measurement. Lines 368 – 371: Even if many of the results are negative? Table 1 should include the units/scales where appropriate and precision regarding the measurements. What does the income refer to? Consider adding the US income in £ as well so we can see how the cohorts compare. From the table, it is difficult to understand how the UK and the US sample compare since only the skewness within each cohort is presented. Table 2 should mention that the estimates are for 1SD change for the exposures Figure 2 the results for the interaction are difficult to see, perhaps it would help to present men and women on different panels? Appendix 1: See previous comment for the methods. Also, more details are needed (i.e. units, scales). Why wasn't income included in the youth SED for the UK?
--	---

VERSION 1 – AUTHOR RESPONSE

Reviewer: 1

Reviewer Name: Leah Li

Institution and Country: UCL Institute of Child Health

Please state any competing interests or state 'None declared': None declared

Please leave your comments for the authors below

The authors investigated the associations of cognitive and sociodemographic factors in early-life with trajectories of adult BMI over three decades in the US National Longitudinal Study of Youth 1979 (NLSY) and the UK National Child Development Study (NCDS).

I have several points for the authors:

1. The introduction could be shortened and focus more on the research question. There is little mention on why they conducted this research using data from these two countries.

Response: We have shortened the text by taking the opportunity to move all specifics on the samples into the methods, at the request of reviewer 2. This has focused our introduction, whilst we have also added a short justification of using the UK and the US.

2. Page 8 lines 152-53: 'BMI and income were assessed at 4 times...' should be 'BMI was assessed at 4 times...'.

Response: We have adjusted this sentence and the next in line with this suggestion.

3. Latent Growth Curve Modeling – line 163 and elsewhere, the authors stated 'The BMI latent variables included variables for overall BMI level, linear slope, and quadratic slope'. I assume the authors meant intercept, slope and the coefficient for the quadratic term of age for BMI. It is not clear what age the overall BMI level is referred to.

Response: Level refers to the same latent variable as intercept. We have added text to clarify that this is the case, and also changed 'slope' to 'change', as in 'linear change', to help clarify.

4. Line 165-66: 'Youth SED, education, sex, cognitive function, and the interaction between sex and 166 cognitive function were regressed on each of these BMI latent variables'. In fact BMI was regressed on Youth SED, education, sex, cognitive function, not the other way around.

Response: Thank you for pointing this out, we have fixed the text.

5. Page 195-199: How did the authors conclude that the rate of BMI increase decreased over time? This can only be obtained from the formula of the growth curve (i.e. the coefficient for the quadratic age term is negative).

Response: Yes, the estimate for the overall quadratic age term is negative. This is in Table S1 as referenced, although it is also apparent from the visual representation of the growth curves in Figure 2.

6. Page 210: '4 to 5 BMI points' should be '4 to 5 kg/m²'.

Response: We have adjusted this as suggested.

7. A general point: for a maximum of 4 BMI measurements per person in the two cohorts, the authors used a very complicated model, with many latent variables – i.e. random intercept, random slope, and random coefficient for age², as well interaction of each factor (e.g. sex, cognition, SEP, education) with these random effects, resulted in a large number of estimates for coefficients and variance/covariance. In table 2, the authors included all the interaction terms with age and age² that are not significant. If they exclude all the interactions with age² that are not significant from the model, some interaction terms with age might become significant. Similarly, if they further remove the interactions with age, some main effects might be significant. Also for these interactions, fixed (not random) effects for age and age² might be sufficient and therefore the models would be a lot simpler and findings might be different. This is only a suggestion.

Response: Thank you for the suggestion. We apologise for any confusion we may have caused in our description of the growth curve models. The model is not a random effects model; rather, the latent variables for BMI are weighted compositions of the BMI measurements, which is standard practice for

*creating intercepts and slope latent variables for growth curve models within a structural equation modelling framework. The only proper interaction we included was sex * cognitive function, and that was because it was a key part of our research question. Following comments from reviewer 2, we have reorganized the methods and beginning of the results, and have kept this comment in mind while doing so. We hope the revised version is clearer.*

Reviewer: 2

Reviewer Name: Veronique Gingras

Institution and Country: Postdoctoral fellow, Harvard Pilgrim Health Care Institute and Harvard Medical School, United States

Please state any competing interests or state 'None declared': None declared

Please leave your comments for the authors below

Overall comments

This is an interesting study with a novel research question. The methods used to assess BMI trajectories are also interesting. Yet, there are some limitations for the interpretation of these findings and clarifications are needed.

First, the authors state in the title and throughout the manuscript that they are looking at early-life cognitive and sociodemographic factors and their relationship with overweight and obesity. However, cognitive function was assessed between 15 – 22 years in one cohort, and at age 11 in the second cohort. SED appears to have been measured in adolescence or young adulthood as well? Perhaps it would be more adequate to change early-life for adolescence or youth?

Response: To be more consistent, we have changed 'early-life' to 'youth' throughout.

Also, there is missing information in the methods, and it makes this study harder to interpret. There seem to be many differences between the two cohorts, and it is unclear whether they can really be compared. For example, the authors give very little information regarding the cognitive function test, which is not the same in both cohorts, and was not assessed at the same time. This is the main exposure variable, I think it would be best to include it in the main manuscript rather than in an appendix, it would be easier to read. Same for education level and SED which are not assessed the same way in both cohorts. Also, is cognitive function assessed at 11 vs. 15-22-year-old comparable? How does this difference affect the results? Authors refer to IQ in two places in the manuscript, was there a specific IQ test performed or does this refer to the cognitive test results?

Response: We have moved information on the samples' variables out of the supplement and into the main text. Please see several comments below which also touch on these issues, particularly our response to the comments regarding lines 298 to 306. The bottom line is that these cognitive tests have been shown to have strong internal and external validity, and we allow for age as a covariate in the NLSY analyses to account for the differences with that sample. The cognitive tests scores are also quite stable across this range of ages, so the differences introduced should be minimal.

The use of IQ in some places was an error, many should have just read as cognitive function, as they refer to the same construct for cognitive function as in every other place in the manuscript. We have changed this. Where we refer to "15 IQ point (1SD)" or use similar language, we do so because of the specific norms regarding IQ having this distribution. Our cognitive function tests can be interpreted in this way, so we have included these statements to aid comprehension for those with more educational or differential psychological background.

Specific comments

Title: See previous comment regarding “early-life”

Response: This has been changed.

Abstract:

Lines 32 – 33: What do authors mean by “most of both studies completed a cognitive function test in youth”? The information regarding the exposure variable (how it was assessed, when and in how many participants) should maybe be part of a sub-section “exposure”?

Response: We agree and we have now included more on the cognitive tests in this revision and changed the section names as suggested.

Line 36: Consider replacing “early-life” and mention the age range when it was measured

Response: We agree and we have replaced ‘early-life’ with ‘youth’. We feel it is best to present the specific age information about these variables in the main text and not the abstract.

Line 36: Socioeconomic disadvantage: Could authors briefly define?

Response: We have added a couple of common examples in the text: “e.g. parental occupational class and time spent in school...”

Line 37: Adulthood – specify which period

Response: We have shifted text from the next section to here to more immediately specific the ages.

Line 43: change “cultures” for “cohorts”

Response: Done.

Line 44: The part of the sentence “a ~ 0.2 ” could be removed since we have this information with the β coefficient

Response: This has been removed.

Lines 44-45: With the β coefficients, add the unit (kg/m²): -0.20, 95% CI -0.33; -0.06 kg/m²; also, the p values are not needed considering we have the CI which doesn’t cross the null; Same for the results on lines 46, 49, 50

Response: We have made the requested changes.

Line 47: change “more than half a point lower BMI” for “lower BMI”, we already have the value with the β coefficient.

Response: We have changed the text as suggested.

Lines 50 – 51: Add the “timeline”: Overall, no cognitive or socioeconomic factors in youth were associated with longitudinal changes in BMI throughout adulthood.

Response: We have changed the text as suggested.

Lines 53 – 55: Perhaps the conclusion should be focused on the primary outcome? What do authors mean by “some average weight pattern”?

Response: By this we mean that the majority of associations are with the overall BMI level (intercept), not gains, i.e. linear or quadratic change. We have changed the text to clarify this: “While sociodemographic and particularly cognitive factors can explain some patterns in individuals’ overall weight levels, differences in who gains weight in adulthood could not be explained by any of these factors.”

Strengths and limitations:

Consider mentioning differences in assessment methods

Line 67: Is “cross-culturally” appropriate here? Change for “in two different populations”?

Response: We think it is accurate to say this is across cultures, though we have added text just before this to be explicit about which cultures, which we think improves this statement.

Lines 73-74: Considering the null results, is it appropriate to mention mechanisms? Maybe change to mention that more studies are needed to clarify if and how cognitive function and SED are linked to obesity

Response: We have adjusted the text as suggested.

Introduction:

Overall, the introduction is quite long. For example, the sentence from lines 80 – 83 might be unnecessary. It is known that obesity is related to poor health outcomes, and this study focuses on BMI itself, not associated co-morbidities.

Response: We wished to highlight the impacts of obesity and, as there is ongoing debate on whether BMI growth is the cause or result of mechanisms that impact health, we judge that it is important to state these associative links.

Line 94: Is it really established that it “precedes obesity”? Has the causal effect been studied, or would it be more appropriate to mention that cognitive function has been found to be linked to later obesity?

Response: By using ‘precede’ we meant that there is a temporal relationship, but not necessarily a causal one. Due to the confusion this might cause, we have changed the text: “...lower cognitive function is linked to obesity later in life.”

Lines 97-98: It is probably an overstatement to say that using an earlier cognitive function measurement avoids confounding. Consider changing for limits confounding and consider moving this sentence to the discussion. Also, can authors be sure that their measurement is pre-morbid? Did

authors exclude from their analysis participants with obesity or complications at the cognitive function assessment?

Response: We have changed the text as suggested, but we wish to keep this in the introduction as many readers would not immediately understand the advantages of having measures of cognitive function in youth.

We unfortunately do not have height or weight measures from before cognitive function tests were given, so we cannot rule out all such individuals that the reviewer mentions. The measure could probably be best describe as “mostly pre-morbid”; for the sake of consistency, we have changed mentions of “pre-morbid cognitive function” to “cognitive function in youth”.

Lines 103-115: This paragraph is quite long and maybe not all information is needed to introduce this study – some of the information in this paragraph would be more appropriate for the discussion

Response: This text is important for us to be able to set up our hypotheses in advance of the discussion. Otherwise, we think many readers would find it arbitrary that we included interactions with sex, which would hinder the overall comprehensibility of the manuscript.

Lines 116-117: “major early-life influences”: change for the specific exposure in this analysis

Response: We have adjusted the text accordingly.

Lines 120 – 127: Move this information to the methods?

Response: This has (in large part) been done.

Methods:

Line 134: What does “- 7506 individuals” represent?

Response: The total number of non-Black non-Hispanic participants. We have clarified this in the text: “... a total of 7506 individuals were in this last category, so 5172 NLSY participants were either Black or Hispanic.”

Line 137-140: Is this information needed?

Response: Yes, genetic and other within-group effects can confound and introduce uncertainty that makes the analyses less effective. At the same time, we think it is important to recognize the uniqueness of each sample; this information could be useful for future interpretation.

Line 147: Give proportions instead of numbers

Response: We have added percentages as with our description of the NLSY.

For both the NLSY and the NCDS studies, specify how many participants were included for this analysis. A flow-chart with participants number (baseline and different assessments) in the original cohort vs. included in this study would be helpful. The methods section would be easier to read if reorganized: before variables, authors could describe the exposures, then outcomes, then covariates, and finally statistical analyses. Also, the information in this section is insufficient; it is not ideal to have to go through supplementary material to understand what the exposure is and how it was assessed.

Most of the information in Appendix 1 should be in the main manuscript. There are several other sections where authors could shorten or rephrase the information to leave more room for the methods.

Response: We agree with these points. Along the lines suggested, we have heavily reorganized the methods section and brought over considerably more information from the supplement.

Most of the information on the numbers of participants in each sample at each wave, has been added into the text, or is contained in Table 1, which we also refer the reader to. Due to the maximum likelihood techniques used in the analyses, the number of participants used by the analyses cannot be determined a priori, and we think this table is a better way to convey the information accurately and completely than a flow-chart under these circumstances.

Results:

Lines 183 – 187: This section is difficult to follow. Ethnicity was not available for all participants? Give proportions instead of numbers. Minority ethnic groups were excluded from the analysis? That was not mentioned in the methods and could possibly introduce selection bias. What does this exclusion represent in terms of # of participants? The flow-chart would be helpful to clarify this.

Lines 190 – 194: This information would be best suited for the methods section.

Lines 195 – 205: Again, most of this information belongs with the methods. Consider clarifying the paragraph on BMI trajectories in the methods, and, if needed, adding an appendix with non-essential information regarding the methods for trajectories.

Response: We have moved much of this text from the results to the methods, and adjusted it to fit accordingly. However, the ultimate form of the model is a result of model selection, thus we think that, for the sake of reader interpretability, we need to present these results in such a way that reflects their importance to the following results. The revised results section reflects this. We have also carried out sensitivity analyses with the minority ethnic groups in response to another comment – please see more on this further along.

Lines 210 – 215: consider changing BMI point for BMI unit – same for throughout the manuscript

Response: We have made this change throughout the manuscript.

Lines 215 – 219: Was IQ the measure of cognitive function?

Response: Where we refer to “15 IQ point (1SD)” or use similar language, we do so because of the specific norms regarding IQ having this distribution. Our cognitive function tests can be interpreted in this way, so we have included these statements to aid comprehension for those with more educational or differential psychological background. We have changed this in several places – see also earlier replies.

Through the results, authors use “BMI growth”; yet, growth usually refers to height. Consider changing for BMI trajectory, or rate of BMI increase where appropriate.

Response: In accordance with a comment from reviewer 1, we have changed ‘growth’ to ‘change’ (as it may be linear change or quadratic change, depending) and ‘rate of increase’.

Lines 240 – 242: What do authors mean by the UK sample had more power? Were calculations

performed or are they referring to the # of participants alone? Is it possible that the measurements were more or less precise in one of the cohorts? Also, this information would be best suited for the discussion.

Response: We refer to this purely in terms of number of participants, and in this instance, we think the information should be presented here because it is relevant to how virtually all the effect sizes that are to be presented should be considered, relative to one another. It is possible that the BMI measurements were less precise in one sample or other, though we do not have the ability to assess this. We have now noted this limitation in the discussion (5th paragraph).

Discussion:

Overall, the discussion is quite long and could be focused. A lot of the information in the discussion is simply a summary of the findings rather than the interpretation of the observed associations, or lack of associations.

Response: BMJ open recommends that authors begin discussions with a statement on the principal findings of the report - meaning and explanations are supposed to be in a later discussion section – which is why the discussion opens and proceeds the way it does. We would prefer to adhere to these recommendations, if possible.

Line 279: Change “appeared to prevent this” for “were associated with these changes”.

Response: Done.

Lines 285-288: Differences in BMI by sex was not the primary outcome of this study; perhaps this section should be shortened into one sentence mentioning that in this study, most of the change in BMI over time was explained by sex?

Response: The section in question presents, as far as we are aware, a novel result. Whereas these sex differences were not an area of primary interest, modelling BMI latent growth curves across countries and sex etc. provides important context for the interpretation of the other results, and is likely to be of interest to other researchers as well, e.g. those working at the intersection of public health and sociology.

Lines 287-288: What do authors mean?

Response: The BMI distribution for Americans is wider than it is for Britons; this is because of the lower limit placed on BMI by physical constraints. The BMI distribution for Americans has a higher centre, which allows the roughly normal BMI distribution(s) to spread out more than it can for Britons. We have added text here to clarify this.

Lines 289 – 297: Authors discuss differences in BMI change from the US vs the UK cohort in the results, and again in the discussion; yet, is this a primary focus of this study? Shouldn't the focus be how cognitive function and SED affect BMI differentially in both cohorts? Also, how comparable are the cohorts beyond BMI? There are not that many baseline variables to understand how different the cohorts are.

Response: As mentioned in the comment above, we think this information is pertinent to reader comprehension of the full complement of results, and the information will be, in its own right, of interest to certain readers.

Lines 298 to 306: IQ or cognitive function? Why present the difference for 15 points in both cohorts when the tests and probably the scales are not the same? Maybe present what 1SD change in cognitive function represents in terms of weight in both cohorts, and specify how many points this 1SD change represents?

Response: In the intelligence literature it is standard to convert cognitive ability tests to this IQ scale, where 1SD is represented by 15 points, and the mean is at 100. This is possible because of the “indifference of the indicator” – same broad factor of cognitive function can be identified from different test batteries (see Stauffer et al., 1996, doi:10.1080/00221309.1996.9921272; Johnson et al., 2004, doi:10.1016/S0160-2896(03)00062-X; Deary, 2012, doi:10.1146/annurev-psych-120710-100353). We think this is the most universal way to present the findings at this point in the discussion, that is, in terms of 1SDs and the equivalent 15 IQ points. We have adjusted the text to clarify this equivalence. Please see previous comments on IQ as well.

Lines 311 – 314: With more participants? The # of participants is already quite high. Could it rather be the precision of the tool that lead to a larger CI?

Response: Yes, with more participants – we have clarified our meaning, and that this is a possibility. The effects sizes are not huge and precision and variability are issues, so these factors may well have contributed to the larger CI.

Lines 323-325: As previously mentioned, this was not indicated in the methods. Why include only participants of European descent? Authors should consider/discuss how this exclusion could introduce selection bias and sensitivity analyses including all participants should be presented.

Response: In addition to cultural factors, there are genetic factors that play a role in overweight and obesity development. Although it is not a perfect technique, it is common to remove minority ethnic groups as a control mechanism when one wishes to specifically consider the culture aspects. We have expanded on this in the revised text.

We have also now carried out additional sensitivity analyses, and as requested, we now present models fit to both samples, including all participants. Please see the next response where we continue on this thread.

Lines 325 – 331: This is unclear. Why do authors believe that their results could be applicable to other ethnic groups if only European descent participants are included?

Response: We think this because of previous research with the NLSY sample in particular, which used all ethnic groups and similar variables, and found few differences between ethnic groups with respect to these variables.

However, with the new sensitivity analyses in hand, we can confirm that including all participants across ethnic groups does not eliminate any of the associations we found. This better supports the idea that these findings can be applied more widely. We have added text to reflect this into the most recent revision.

Lines 331-333: The number of participants is already quite large, see previous comment regarding precision of the tools vs. # of participants for CI precision

Response: We take the reviewer's point, and have noted the issues of precision. Our text here is more measured – we do not claim that we missed out on finding effects, we just wish to acknowledge that power in the British sample was greater.

Lines 334 – 335: At which time points? Different assessment timing from the current analysis?

Response: The time points were the same; we have clarified this in the revised text.

Lines 344 – 345: Even though results were different in the two cohorts?

Response: By this we meant that the effects that were replicated across the sample can be viewed as more robust. We have clarified this in the revision.

Lines 346 – 356: I do not quite understand this paragraph. Why are authors discussing genetic associations? Isn't this paragraph out of the scope of this analysis?

Response: It is common in discussion sections to consider the possible mechanisms of associations that have been found, even when one has not looked at all contributing mechanisms in the actual study. In this case, we think it is relevant to consider the mechanisms of the associations, which would include genetic factors, as well as environmental and psychological factors we directly analyse. This is important because it can help point to further implications and direct future research. For instance, we consider such an implication with our discussion of "phenocopying", which is situated in a mixed behavioural and genetic context.

Lines 357 – 363: The fact that associations were only found with overall BMI and not BMI change over time does not really support the importance of overall BMI. BMI is not a perfect method to assess obesity. Several studies showed that BMI does not always reflect adiposity and cardiometabolic risk, authors could mention this limitation of this measurement.

Response: We have added this limitation of using BMI - earlier on in the manuscript, where we discuss other limitations.

Lines 368 – 371: Even if many of the results are negative?

Response: We wished to highlight the positive results, i.e. that there were notable influences of cognitive function in both samples, though there are very likely mechanistic differences given the interaction with sex. We do think that, if real-world solutions are to grapple with psychological correlates (and perhaps causes) of disease, then both the positive and null results must be carefully considered, but in this study we do not have the ability to state that absence of evidence is evidence of negative results.

Table 1 should include the units/scales where appropriate and precision regarding the measurements. What does the income refer to? Consider adding the US income in £ as well so we can see how the cohorts compare. From the table, it is difficult to understand how the UK and the US sample compare since only the skewness within each cohort is presented.

Response: We have clarified and added units where appropriate, though many are standardized unitless variables; this is referred to in the caption. E.g. precision is given for all variables with their SDs, but not for the standardized variables.

We do not have the information (specifically the precise dates of data collection) necessary to accurately convert USD to GBP or GBP to USD due to the historical nature of these data. Moreover, both the US and UK income figures have been carefully computed by the teams managing these cohorts, and we would prefer not to violate any of their assumptions or muddy the interpretability of their work by adjusting the income values. Comparing these values is an econometric question that is beyond our expertise and we think the scope of this paper, as well.

Table 2 should mention that the estimates are for 1SD change for the exposures

Response: We have added this into the table caption.

Figure 2 the results for the interaction are difficult to see, perhaps it would help to present men and women on different panels?

Response: We appreciate the reviewer’s attention to the details of these plots. We have also tried to draw these plots in several different ways (including the way suggested) and we determined amongst ourselves that the current plots are the best way to present these curves

Appendix 1: See previous comment for the methods. Also, more details are needed (i.e. units, scales). Why wasn’t income included in the youth SED for the UK?

Response: We have moved much of the information in the supplement into the main methods and made according adjustments, as requested. In the case of UK youth SED, income was not recorded at this early wave. Occupational social class was collected far more frequently during this era; income data was not all that commonly collected, unfortunately.

VERSION 2 – REVIEW

REVIEWER	Véronique Gingras Harvard Medical School and Harvard Pilgrim Health Care Institute, United States
REVIEW RETURNED	07-Nov-2019

GENERAL COMMENTS	With this revision, authors have improved the manuscript and provided enough information to explain why they made some of the decisions regarding their manuscript. In particular, the introduction is clearer and more focused, and the methods are much easier to read with the reorganization. Although I still find the discussion a little descriptive and long, the authors do provide some perspective and insight, and thus there are no additional comments. One minor comment: Authors changed “early life” to “youth” for most of the manuscript, and left “early life” in some instances, which is okay considering the context. However, in the aim of the abstract, I would suggest changing early life for youth as well.
---